# Increased Lipid Peroxidation and Lowered Antioxidant Defenses Predict Methamphetamine Induced Psychosis

**DOI:** 10.3390/cells11223694

**Published:** 2022-11-21

**Authors:** Hussein Kadhem Al-Hakeim, Mazin Fadhil Altufaili, Abbas F. Almulla, Shatha Rouf Moustafa, Michael Maes

**Affiliations:** 1Department of Chemistry, College of Science, University of Kufa, Kufa 54002, Iraq; 2Department of Psychiatry, Faculty of Medicine, King Chulalongkorn Memorial Hospital, Chulalongkorn University, Bangkok 10330, Thailand; 3Medical Laboratory Technology Department, College of Medical Technology, The Islamic University, Najaf 54001, Iraq; 4Clinical Analysis Department, College of Pharmacy, Hawler Medical University, Erbil 44001, Iraq; 5Department of Psychiatry, Medical University of Plovdiv, 4000 Plovdiv, Bulgaria; 6School of Medicine, Barwon Health, IMPACT, The Institute for Mental and Physical Health and Clinical Translation, Deakin University, Geelong VIC 3216, Australia

**Keywords:** psychosis, oxidative and nitrosative stress, antioxidants, neurotoxicity, schizophrenia

## Abstract

Background: a significant percentage of methamphetamine (MA) dependent patients develop psychosis. The associations between oxidative pathways and MA-induced psychosis (MIP) are not well delineated. Objective: the aim of this study is to delineate whether acute MA intoxication in MA dependent patients is accompanied by increased nitro-oxidative stress and whether the latter is associated with MIP. Method: we recruited 30 healthy younger males and 60 acutely intoxicated males with MA dependence and assessed severity of MA use and dependence and psychotic symptoms during intoxication, and serum oxidative toxicity (OSTOX) biomarkers including oxidized high (oxHDL) and low (oxLDL)-density lipoprotein, myeloperoxidase (MPO), malondialdehyde (MDA), and nitric oxide (NO), and antioxidant defenses (ANTIOX) including HDL-cholesterol, zinc, glutathione peroxidase (GPx), total antioxidant capacity (TAC), and catalase-1. Results: a large part (50%, n = 30) of patients with MA dependence could be allocated to a cluster characterized by high psychosis ratings including delusions, suspiciousness, conceptual disorganization and difficulties abstract thinking and an increased OSTOX/ANTIOX ratio. Partial Least Squares analysis showed that 29.9% of the variance in MIP severity (a first factor extracted from psychosis, hostility, excitation, mannerism, and formal thought disorder scores) was explained by HDL, TAC and zinc (all inversely) and oxLDL (positively). MA dependence and dosing explained together 44.7% of the variance in the OSTOX/ANTIOX ratio. Conclusions: MA dependence and intoxication are associated with increased oxidative stress and lowered antioxidant defenses, both of which increase risk of MIP during acute intoxication. MA dependence is accompanied by increased atherogenicity due to lowered HDL and increased oxLDL and oxHDL.

## 1. Introduction

Methamphetamine (MA), a potent psychostimulant derivative of amphetamine, is the second-most misused substance after cannabis, and is a worldwide health concern because of its ubiquity, high prevalence, and rising overdose-related death rates [1,2]. In the United States, MA use increases among individuals who use other drugs, including heroin, from 22.5% to 46.7% in one year [3]. People usually use MA to cope with their weariness, induce a state of pleasure, facilitate social interaction, enhance libido, boost productivity at home and work, and lose weight by decreasing appetite [4,5]. Nevertheless, several adverse consequences are associated with using MA, one of the most important being addiction [6,7]. MA may trigger many neuropsychiatric symptoms and behaviors among users, including a wide spectrum of affective, cognitive, somatic, psychotic, and behavioral manifestations, including violent behavior, insomnia, and irritability [8,9,10,11,12]. Consequently, these symptoms may cause progressive social and occupational decline [13,14].

Many MA-dependent individuals may experience new-onset psychotic symptoms or a worsening of their psychotic symptoms [15]. Auditory and tactile hallucinations, paranoid delusions, and ideas of reference are repeatedly reported as prominent MA-related psychotic symptoms [16,17]. Previous studies indicate a wide prevalence range of MA-induced psychosis (MIP) ranging from 7% [17] to 76% [18]. A meta-analysis revealed that the prevalence of MA-related psychotic disorders is 36.5%, with a lifetime prevalence of 42.7% [19]. Nevertheless, MIP is a challenging concept because the diagnostic criteria are not well-defined and because various etiologic and pathophysiological factors are associated with MIP. Patients with a genetic predisposition to psychosis and those who have already suffered from a psychotic disorder like schizophrenia are more likely to develop MA-associated psychotic symptoms [17,20]. Important risk factors for MIP are increased MA use, higher dependence, and frequent intake [21].

Schizophrenia and MA dependence and MIP share several clinical characteristics, and MIP is frequently considered to be a mechanistic model of schizophrenia [22,23,24,25]. Dopaminergic signaling in the mesocorticolimbic and nigrostriatal networks has been implicated in schizophrenia and MIP [2]. The same disorders in long interspersed nuclear element-1 (LINE 1) partial methylation patterns are detected in MIP and paranoid schizophrenia and are more pronounced in the latter [24,25]. Schizophrenia may result from neurotoxic processes [26] and MA usage has neurotoxic effects on cortical interneurons [27]. Activated immune-inflammatory and neuro-oxidative stress play a role not only in schizophrenia or schizophrenia phenotypes [26,28,29], but also in MA dependence [25,30,31]. High levels of nitro-oxidative stress (NOS) are confirmed in schizophrenia as indicated by increased reactive oxygen (ROS) and nitrogen species (RNS), increased lipid peroxidation as indicated by increased levels of lipid hydroperoxides, and increased protein oxidation as indicated by increased advanced oxidation products (AOPP), and lowered total antioxidant defenses [32,33,34,35,36]. Oxidative stress biomarkers, including those indicating oxidative damage end-products, oxidant enzyme activities, and lowered antioxidant levels are also observed in MA dependent patients [37,38].

Therefore, we hypothesized that indicants of increased oxidative stress toxicity (OSTOX) and decreased antioxidant defenses (ANTIOX) may be detected in patients with MA dependence and MIP during MA intoxication. Nonetheless, no studies have reported associations between OSTOX/ANTIOX and MIP in MA-dependent individuals during MA intoxication. Hence, the aim of the present study is to examine whether MA dependence and MIP during intoxication are characterized by (a) increased serum NOS/OSTOX biomarkers, including malondialdehyde (MDA), myeloperoxidase (MPO), nitric oxide (NO), oxidized high-density lipoprotein (oxHDL) and low-density lipoprotein (oxLDL) levels; and (b) lowered ANTIOX biomarkers, including catalase-1, glutathione peroxidase (Gpx), total antioxidant capacity (TAC), HDL cholesterol, and zinc. The data are analyzed using a precision nomothetic approach [39].

## 2. Material and Methods

### 2.1. Participants

In the present study, we recruited 60 MA-intoxicated male patients with MA substance use disorder (SUD) at the Psychiatry Unit, Al-Hussein Medical City, Kerbala Governorate, Iraq, from April 2022 to August 2022. The patients were diagnosed according to the Diagnostic and Statistical Manual of Mental Disorders (5th edition) (DSM-5) as moderate to severe SUD [40]. Due to the religious state of Karbala city, we could not recruit female MA SUD patients and included male SUD patients only. All SUD patients started MA intake before at least three months prior to the study. Prior to hospitalization and blood sampling, no antipsychotic medications were administered to any of the patients. All patients were admitted to the hospital for the first time. In patients, a urine examination showed a positive MA test. Patients were excluded if they showed a lifetime or current diagnosis of other axis-1 diagnoses including mood disorders, schizophrenia, schizo-affective psychoses, obsessive compulsive disorder, post-traumatic stress disorder, generalized anxiety disorder, panic disorder, autism spectrum disorders. In this case, 30 apparently healthy controls, namely family and friends of staff or friends of patients served as controls. None of the controls had ever taken any psychoactive drugs (except tobacco use disorder) and none showed a current or lifetime DSM-5 axis I diagnosis of SUD, psychosis or schizophrenia, bipolar disorder or other affective disorders, or a family history in first-degree relatives of schizophrenia or psychosis. The patients and controls were excluded if they had ever taken immunosuppressive treatments, glucocorticoids or antipsychotic agents or had been diagnosed with a neurodegenerative or neuroinflammatory illness such as Alzheimer’s disease, Parkinson’s disease, multiple sclerosis, or stroke. Additionally, individuals with (auto)immune diseases such as inflammatory bowel disease, rheumatoid arthritis, COPD, psoriasis, or diabetes mellitus were excluded.

The study followed Iraqi and international privacy and ethics laws. Before participating in this study, all participants, and their guardians (legal representatives of the patients are mother, father, brother, spouse, or son) gave written informed consent. The study was approved by the ethics committee (IRB) of the College of Science, University of Kufa, Iraq (89/2022), Karbala Health Directorate-Training and Human Development Center (Document No.18378/2021), which follows the Declaration of Helsinki’s International Guideline for Human Research Protection.

### 2.2. Clinical Assessments

In order to collect patient and control data, a senior psychiatrist with expertise in addiction conducted a semi-structured interview and scored rating scales to assess severity of MA dependence, use and psychosis. The Severity of the Dependence Scale (SDS) was used to estimate the severity of MA dependence, namely 5 items: (a) did you ever think your use of MA was out of control, (b) did the prospect of missing MA make you very anxious or worried, (c) did you worry about your use of MA, (d) did you wish you could stop, and (e) how difficult would you find it to stop or go without MA [41]. We also registered age at onset, duration of MA dependence, daily dosage (grams), route of administration (ordinal variable with no = 0, orally ingested = 1, smoked or snorted = 2, and injected = 3), number of previous psychotic episodes and days hospitalized after admission for acute intoxication. We also registered lifetime cannabis and alcohol use as well as cannabis and alcohol dependence. The patients did not use any other drugs of dependence, including opioids, cocaine, or heroin. The same day, we assessed the Brief Psychiatric Rating Scale (BPRS) [42] and the Positive and Negative Syndrome Scale (PANSS) [43] rating scales. In analogy with our studies in schizophrenia [44,45,46], we computed different z unit-weighted composite scores based on the items of the BPRS and PANNS to reflect the severity of MIP, namely: (a) psychosis as sum of the z transformations of hallucinations (BPRS) + suspiciousness (BPRS) + delusions (PANSP1) + hallucinatory behavior (PANSS P3) + suspiciousness (PANSS P6), (b) hostility was computed as the sum of z transformations of hostility (BPRS) + uncooperativeness (BPRS) + hostility (PANSS P7) + uncooperativeness (PANSS G8) + poor impulse control (PANSS G14), (c) excitement: sum of z transformations of grandiosity (BPRS) + excitement (BPRS) + excitement (PANSS P4) + grandiosity (PANSS P5), (d) mannerism and posturing (BPRS) + mannerism and posturing (PANSS G5), and (e) formal thought disorders (FTD): sum of z transformations of conceptual disorganization (BPRS) + unusual thoughts (BPRS) + conceptual disorganization (PANSS P2) + difficulties in abstract thinking (PANSS N5) + stereotyped thinking (PANSS N7). Here, we use PHEM symptoms to denote psychosis, hostility, excitation and mannerism [45,46,47]. In addition, we computed, post-hoc, a new index based on the most prominent MA intoxication associated psychotic (MAI) symptoms in our MA dependent patients, namely delusions (PANSS P1) + conceptual disorganization (PANSS P2) + suspiciousness (PANSS P6) + difficulty in abstract thinking (PANSS N5). Tobacco use disorder (TUD) was diagnosed following DSM-5 criteria. The following formula was used to compute the body mass index (BMI): body weight (kg)/length (m^2^).

### 2.3. Biomarkers Assays

Fasting venous blood was obtained from all participants in the early morning hours after awakening and before having breakfast. After 15 min at room temperature, the blood was allowed to coagulate for 10 min before being centrifuged at 3000 rpm for 10 min. The separated serum was then transferred to Eppendorf tubes and stored at −80 °C until analysis. A urine MA test was carried out immediately after admitting the acutely intoxicated patient using the urine Multi-Drug 12 Drugs Rapid Test Panel kit supplied by Citest Diagnostics Inc. (Vancouver, Canada). Serum zinc was measured spectrophotometrically using a ready-for-use kit supplied by Agappe Diagnostics^®^ (Cham, Switzerland). HDL was measured using a kit supplied by Spinreact^®^ (Gerona, Spain) based on a direct method. The serum levels of catalase, Gpx, MPO, MDA, oxHDL, oxLDL, TAC, and NO were measured using commercial ELISA kits supplied by Nanjing Pars Biochem Co. Ltd. (Nanjing, China). All kits were based on a sandwich technique. The procedures were followed according to the manufacturer’s instructions without any modifications. The intra-assay coefficients of variation (CV) for all the assays were <10.0% (precision within-assay). Consequently, we computed 3 composite scores: (a) oxidative stress toxicity (OSTOX) as the sum of the z transformation of MPO (zMPO) + zMDA + zoxHDL + zoxLDL, and (b) antioxidant defenses (ANTIOX): zcatalase + zGpx + zTAC + zZinc + zHDL; and (c) the OSTOX/ANTIOX ratio as zOSTOX–zANTIOX.

### 2.4. Statistical Analysis

Analysis of variance (ANOVA) was employed to examine differences between groups in continuous variables and analysis of contingency tables (χ2-test) to investigate the association between nominal variables. Pearson’s product-moment correlation coefficient was employed to examine the correlation between two scale variables. We utilized a multivariate general linear model (GLM) in order to delineate the associations between study group (healthy control and patient groups) and the psychiatric rating scale scores composites and biomarkers, while controlling for confounding variables, namely age, sex, smoking, and education. We calculated the estimated marginal mean values (SE) and employed protected (namely: the omnibus test is significant) least significant difference (LSD) tests to carry out pairwise comparisons among the group means. We additionally applied false discovery rate (FDR) p-correction to the multiple comparisons [47]. In addition, multiple regression analysis has been utilized to examine whether the biomarkers can significantly predict the various symptom domains. We also used a stepwise automated approach with a *p*-value of 0.05 for entry and 0.06 for removal from the model. Standardized beta coefficients with t statistics and exact *p*-value were computed for each of the predictors’ variables and we also compute the model statistics (F, df and *p* values) and total variance explained (R^2^) as effect size. Furthermore, we used the variance inflation factor (VIF) and tolerance to examine collinearity and multicollinearity issues. We tested for heteroskedasticity using the White and modified Breusch-Pagan homoscedasticity tests and, if necessary, utilized univariate GLM analysis to estimate parameters with substantial error margins. Two-tailed tests were used to evaluate the significance, set at *p* = 0.05 in SPSS version 28 (windows) to perform all statistics.

Partial Least Squares (PLS) analysis was conducted to delineate the causative associations among MA dependence, alterations in NOS biomarkers, and the symptom domain scores induced by MA. Toward this end, we examined if one validated latent vector could be extracted from psychosis, hostility, excitement, and formal thought disorders and, if so, used this factor as output variable. The biomarker input variables were entered as single indicators and the common input variable was a latent vector reflecting MA-dependence and use. We perform complete PLS analysis when the following criteria are met: (a) all loadings on the latent vectors should be >0.6 at *p* < 0.001, (b) adequate construct and convergence validity as indicated by rho A > 0.8, Cronbach’s alpha >0.7, composite reliability >0.7, and average variance extracted (AVE) > 0.5, (c) construct’s cross-validated redundancy should be sufficient as indicated by blindfolding analysis, (d) the results of the Confirmatory Tetrad Analysis (CTA) should display that the latent vectors constructed are correctly described as a reflective model, (e) PLSpredict analysis should prove that prediction performance of the model is efficient, and f) adequate model fit as indicated by standardized root squared residual (SRMR) values < 0.08. Once the quality of the model has been confirmed based on the criteria mentioned above, we carried out a complete PLS-SEM pathway analysis using 5000 bootstraps to compute the path coefficients (with *p*-values) along with specific and total indirect (mediated) effects and total effects. A priori power analysis shows that the estimated sample size for a PLS analysis (which is the primary analysis) performed using a power = 0.8, alpha = 0.05, effect size = 0.17 and using maximal 5 predictors should be 82 participants. Accordingly, we included 90 subjects in the present study. Principal component (PC) analysis was performed to extract the first PC from interrelated variables using SPSS version 28. The factorability was checked with the Kaiser-Meyer-Olkin (KMO) metric and the Bartlett’s chi-square test.

## 3. Results

### 3.1. Cluster and Factor Analysis

We performed a two-step cluster analysis to divide the patients into two groups according to the MAI symptoms, OSTOX, ANTIOX and OSTOX/ANTIOX, while we entered MA dependent patients versus controls as categorical variable. Three clusters were formed with a silhouette measure of cohesion and separation of 0.62. These included healthy controls (n = 30) and individuals with lower psychotic symptoms and oxidative stress (MA-PSO, n = 30) versus those with high psychotic symptoms and oxidative stress (MA+PSO, n = 30). We were able to extract validated PCs from SDS1 (loading = 0.913), SDS2 (0.951), SDS4 (and 0.691), and SDS5 (0.878) (KMO = 0.832, Bartlett’s test chi-square = 287.09, df = 6, *p* < 0.001, AVE = 0.747, labeled PC_SDS). We were also able to extract a validated PC from PC_SDS (0.959), dosage (0.854), MA use last month (0.961) and route of administration entered as an ordinal variable (0.672) (KMO = 0.743, Bartlett’s test chi-square = 367.85, df = 6, *p* < 0.001, AVE = 0.756, labelled: PC_MA).

### 3.2. Sociodemographic Data and MA Features in the Study Groups

The sociodemographic characteristics of controls and both MA subgroups are presented in Table 1. The results show that MA+PSO patients are older and show a higher unemployment rate and PC_SDS and PC-MA scores than MA-PSO patients, whilst there are no significant differences in BMI, education, and marital state. Other differences between both MA groups are a higher rate of injections of abused MA, duration of MA dependence, and dosing of MA in the MA+PSO group than in the MA-PSO group, whilst there were no differences in number of MIP episodes, duration of index MIP episode, and days admitted to hospital due to MA intoxication. MA abusing patients show more TUD than controls but no differences in alcohol dependence and current intake or lifetime cannabis use. None of the patients or controls showed any abuse of other illicit drugs including cocaine, heroin, or opioids.

### 3.3. Psychotic Symptoms Scores among Study Groups

Table 2 shows the measurements of MIP associated symptoms, namely MAI symptoms, psychosis, hostility, excitement, mannerism and FTD, in the three study groups. There were significant differences in all MIP-associated domains between MA-PSO and MA+PSO, except in mannerism. These results remained significant after FDR correction.

### 3.4. Serum Biomarkers Levels among the Study Groups

The measured biomarkers are presented in Table 3. The results show that Gpx, NO, and zinc are significantly decreased in MA+PSO as compared with controls. MDA, oxLDL and OSTOX (all three increased) and HDL (decreased) were significantly different between MA patients and controls. TAC and ANTIOX and the OSTOX/ANTIOX ratio were significantly different between the three study groups, with TAC and ANTIOX decreasing and OSTOX/ANTIOX increasing from controls → MA-PSO → MA+PSO. In this case, oxHDL was significantly higher in the MA+PSO group than in controls.

We also computed the differences in biomarkers between MA dependent patients and controls. We found that MDA, oxLDL OSTOX and OSTOX/ANTIOX ratio (all *p* < 0.001) and oxHDL (*p* = 0.012) were significantly higher in MA dependence than in controls, while catalase (*p* = 0.033), HDL (*p* = 0.005), TAC, zinc and ANTIOX (all *p* < 0.001) were significantly lower in MA dependence than in controls. FDR p correction did not change any of these results.

### 3.5. Intercorrelation between PC_SDS, PC_MA, Biomarkers, and Psychotic Symptoms

We performed correlation analyses to delineate the associations between PC_SDS, PC_MA, MA-induced psychotic symptoms, and biomarkers (Table 4). Our results indicate that PC_SDS and PC_MA are significantly associated with all MA-induced symptom domains in both groups combined. Moreover, also in MA patients there were significant correlations between PC_SDS and PC_MA and MAI symptoms, psychosis, excitement and FTD. OSTOX (positively), ANTIOX (inversely) and OSTOX/ANTIOX (positively) were significantly associated with all MIP symptom domains in both controls and patients combined, except ANTIOX which was not significantly associated with mannerism. In MA patients, no significant correlations were detected between the OSTOX and ANTIOX scores and the symptom domains.

### 3.6. Prediction of MIP Symptoms and the OSTOX/ANTIOX Ratio

In Table 5, regression #1, we selected MAI symptoms as a dependent variable and performed multiple regression analysis showing that 35.2% of the variance could be explained by TAC, HDL and zinc (inversely) and oxHDL and oxLDL (positively associated). Forced entry of age, BMI, education, TUD, and alcohol dependence showed that TUD (t = −2.14, *p* = 0.035) was the only significant predictor and that the effect of the biomarkers remained significant. Table 5, regression #2 shows that 30.8% of the variance in the MAI score was explained by the OSTOX/ANTIOX ratio. Figure 1 shows the partial regression of the MAI symptom score on the OSTOX/ANTIOX ratio (adjusted for age, TUD, education, BMI, and alcohol use). Adding TUD showed that both the OSTOX/ANTIOX ratio and TUD were significant predictors and together explained 35.9% of the variance (F = 24.34, df = 2/89, *p* < 0.001), although the impact of OSTOX/ANTIOX (β = 0.479, t = 5.59, *p* < 0.001) was much higher than that of TUD (β = 0.238, t = 2.63, *p* = 0.010). Nevertheless, univariate GLM analysis shows that TUD (F = 0.017, df = 1/84, *p* = 0.896) has no significant effect on the MAI score above and beyond that of the diagnostic classification (F = 66.53, df = 2/84, *p* < 0.001).

The second part of Table 5 shows the results of multiple regression analyses with MA-induced psychosis, hostility, excitement, mannerism and formal thought disorders as dependent variables and OSTOX and ANTIOX biomarkers as explanatory variables, while allowing for the effects of confounders. Table 5, regression #3, indicates that 31.6% of the variance in psychosis was explained by the regression on TAC and zinc (inversely) and oxLDL (positively). Regression #4 shows that TAC (inversely) and oxLDL (positively) could explain 14.7% of the variance in hostility. Regression #5 reveals that 17.5% of the variance in the excitement could be explained by zinc (inversely) and oxLDL and age (both positively). Figure 2 shows the partial regression of the excitement score on serum zinc. Regression #6 indicates that 14.1% of the variance in mannerism was explained by the cumulative effects of oxLDL (positively) and zinc (inversely). A larger part of the variance in FTD (35.5%) was explained by the regression on TAC and zinc (both inversely), oxHDL and MPO (both positively). Figure 3 shows the partial regression of the FTD score on oxHDL.

Multiple regression analysis (Table 6, regression #1) showed that PC_SDS and MA dosing were the most significant predictors of the first PC extracted from the 5 symptom domains and explained 81.8% of its variance. Figure 4 shows the partial regression of MA symptoms on PC_SDS. Multiple regression analysis (Table 6, regression #2) showed that PC_SDS and MA dosing were the most significant predictors of the OSTOX/ANTIOX ratio and explained 44.7% of its variance. Figure 5 shows the partial regression of the OSTOX/ANTIOX ratio on MA dosing. In MA patients, we found that 7.1% of the variance in the OSTOX/ANTIOX ratio was explained by MA dosing (β = 0.267, t = 0.211, *p* = 0.039). We found that 27.1% of the variance in OSTOX was explained by PC_SDS (F = 32.64, df = 1/88, *p* < 0.001), while 21.6% in ANTIOX was explained by MA dosing (F = 24.22, df = 1/88, *p* < 0.001).

### 3.7. Results of PLS Analysis

We used PLS analysis (Figure 6) to delineate whether the impact of MA use and dependence (entered as a latent vector extracted from MA use, MA dosing, route of administration, and PC_SDS) on the MA-related symptoms (entered as a latent vector extracted from five symptom domains (MAI symptoms, psychosis, hostility, excitation and FTD; mannerism did not load highly on this factor and was consequently deleted from the final model) is mediated by increased OSTOX and ANTIOX biomarkers. After feature reduction we found that HDL, oxDL, TAC and zinc were the significant predictors of the MIP symptoms. The quality of the current PLS model was adequate with SRMR = 0.042. The MA symptoms factor showed adequate convergence and construct reliability with AVE = 0.853, rho A = 0.967, composite reliability = 0.970, and Cronbach alpha = 0.957, while all loadings were >0.887 at *p* < 0.001. The PC_MA symptoms factor also showed adequate convergence and construct reliability with AVE = 0.756, rho A = 0.914, composite reliability = 0.924, and Cronbach alpha = 0.886, while all loadings were > 0.665 at *p* < 0.001. CTA confirmed that both latent vectors were not mis-specified as reflective models. PLSPredict showed that the construct indicators Q2 predict values were all > 0 indicating that the prediction error was lower than the naivest benchmark. Complete PLS analysis performed using 5000 bootstraps revealed that 29.9% of the variance in MA symptoms could be explained by the regression on HDL, TAC, and zinc (all inversely) and oxLDL (positively). In addition, PC_MA predicted 10.3% of the variance in HDL, 13.2% in oxLDL, 20.2% in TAC and 15.2% in zinc. As such, PC_MA had significant specific indirect effects on MA symptoms mediated by oxLDL (t = 2.24, *p* = 0.013) and TAC (t = 1.82, *p* = 0.034) as well as significant total effects (t = 5.15, *p* < 0.001).

## 4. Discussion

### 4.1. Clinical Aspects of MA Intoxication and MIP

The first major finding of the current study is that patients with acute intoxication could be divided into two relevant clusters: 50% of all patients could be assigned to a cluster with high scores on all PHEM symptoms and FTD as well as elevated OSTOX/ANTIOX values, and another 50% to a cluster with low symptoms and biomarker scores. In addition, the former group exhibit a greater severity of dependence, a longer duration of dependence, a higher MA dose prior to intoxication and hospitalization, and a greater proportion of MA injection rate. As such, a significant association was found between MA dependence/dosing/route, elevated OSTOX/ANTIOX levels, and psychotic symptoms.

Previous research revealed that 40% of MA users may exhibit positive psychotic symptoms and cognitive symptoms comparable to those found in schizophrenia, supporting that MIP may serve as a model for schizophrenia [27,48,49,50]. Our findings also extend previous reports indicating that higher MA dependence, heavy MA use, increased frequency of use, and higher MA dosing (blood concentrations) are risk factors for MIP (review: [21,51].

MIP can be difficult to diagnose since it might be mistaken for a primary psychotic condition, such as schizophrenia or bipolar disorder, or a psychosis resulting from the use of another substance [40,52]. In this respect, it should be highlighted that patients with premorbid schizophrenia or affective disorders were excluded from the present investigation and that no patients with other major illicit drug use disorders were included. Both MIP (this study) and schizophrenia [46,53,54] are characterized by increased PHEM symptoms and FTD, although none of our patients suffered from clinically relevant hallucinations. However, many patients had high scores on delusions, conceptual disorganization, suspiciousness, excitement, and hostility. Previously it was described that MIP shares many symptomatic similarities with paranoid schizophrenia (e.g., based on DSM-IV-TR criteria) [51].

### 4.2. MA Dependence, OSTOX and ANTIOX

The second major finding of this study is that there are statistically significant differences in NOS biomarkers between patients with MA dependence and healthy controls. These findings extend those of earlier studies indicating that a substantial proportion of MA-dependent patients exhibit oxidative stress and psychosis as a result of their drug use [16,17,25,55]. In addition, we discovered that MA-dependent patients had significantly elevated MDA, oxHDL, oxLDL, OSTOX, and OSTOX/ANTIOX levels, and decreased catalase, TAC, HDL, zinc and ANTIOX levels as compared with controls. In addition, MA dependence coupled with MA dosing largely predicted increased OSTOX/ANTIOX values, with dependence being associated with OSTOX and MA dosing with decreased antioxidant defenses.

These findings extend previous findings that MA use increases MDA and other markers of lipid peroxidation in the blood and brain and decreases catalase, Gpx, GSH, SOD, and thiols groups [56,57,58,59,60,61]. MA also causes mitochondrial oxidative damage in human T lymphocytes [62] and in vivo and in vitro MA exposure increases ROS production in the central nervous system (CNS) [63,64]. Nonetheless, the present results demonstrate that in addition to MDA and Gpx, elevated oxLDL and oxHDL and decreased zinc and HDL are important alterations in MA dependence and abuse, and that it is essential to distinguish between MA dependence and MA dosing/route of administration.

In addition, in rodent models, MA administration induces lipid peroxidation with elevated MDA and hydroxynonenal levels, protein oxidation with elevated protein carbonyl levels, increased NO and nitroprotein production, and decreased levels of antioxidant defenses including superoxide dismutase, catalase, and the glutathione system [65]. Prior animal studies demonstrated that chronic administration of MA causes nerve terminal degeneration in various brain regions, including the cortex, striatum, hippocampus, and olfactory bulb [66,67,68] and that these deleterious effects are mediated by ROS, including hydroxyl radicals, which cause oxidative damage not only to lipids but also to DNA and proteins [66,69]. When such damage cannot be repaired, it can accumulate and lead to cellular dysfunctions, neurotoxicity and neurodegeneration [70,71].

There is growing evidence that the neurotoxicity induced by MA is mediated by molecular pathways including immunological and oxidative processes, epigenetic changes, and changes in neurotransmitter turnover. MA use may cause the sustained release of catecholamines like dopamine [25,72], as well as neurodegeneration in the hippocampus and frontal, prefrontal, and temporal lobes; white matter hypertrophy and gliosis; glutamate neurotoxicity; and damage to neuronal dendrites [73]. MA-induced neurotoxicity is associated with increased production of dopaminergic quinones, increased production of ROS, inflammation and microglial activation, increased glutamatergic activity, activated apoptotic pathways, and activated oxidative pathways [25,74,75]. The enhanced apoptotic pathways in MA-dependent patients, including BECN1, MAP1ALC3, CASP8, TP53, and BAX, may be explained by activated oxidative pathways [76]. Additionally, MA may change the cholinergic anti-inflammatory system and the gut microbiota increasing leaky gut [77]. Leaky gut may then further activate immune-inflammatory pathways and drive oxidative, immune-inflammatory, and related pathways. Additionally, modifications in LINE 1 partial methylation patterns caused by MA may result in changes in oxidative and immune-inflammatory pathways [25]. MA exposure changes the production of not only immune-oxidative and apoptotic pathways but also neurotrophins, including brain-derived neurotrophic factor (BDNF) [30,78]. For example, long-term MA users had a substantial decrease in BDNF in the dorsolateral prefrontal cortex [76]. In animal studies, oxidative stress may decrease BDNF levels, whilst injection of vitamin E may reverse this impact [79]. Importantly, tangled connections are reported between activated immunological and oxidative stress pathways and BDNF in major neuropsychiatric illnesses, with lower BDNF levels being related with diminished neurotrophic protection and subsequently greater neurotoxicity [76,80].

Nevertheless, our investigation revealed that MA dependence, acute intoxication, and MA dosage are accompanied by diminished antioxidant defenses. In this connection, Zare et al. [76] showed a decreased glutathione content in the dorsolateral prefrontal cortex of patients with persistent MA dependence and neurotoxic processes. Toborek et al. discovered that low levels of antioxidant defenses may contribute to the brain-blood-barrier damage induced by high levels of oxidative stress in MA-dependent individuals [81].

It should be underscored that MA is accompanied by detrimental effects on cardiovascular health including atherosclerosis, coronary disease, and stroke [82,83]. Abuse of MA is known to enhance the production of atherosclerotic plaques and the risk of myocardial infarction [82]. MA reduces some traditional risk factors, such as total cholesterol and BMI, but increases risk via effects on ROS and RNS production, proinflammatory genes and responses, and proatherogenic cytokines (including interleukin-1β and tumor necrosis factor-α), chemotaxis, adhesion molecules, endothelial activation, intimal cholesterol deposition, and increased catecholamine production [82,83]. Nonetheless, the present study demonstrates the existence of other and possibly even more significant proatherogenic risk factors, namely decreased HDL and elevated oxLDLand oxHDL levels. Oxidation of LDL is a significant early event in atherosclerotic processes via effects on lipoproteins, hypertension-related mechanisms, pro-inflammatory cytokines, and the production of IgG antibodies directed against oxLDL [84,85,86,87]. Increased production of oxHDL is a significant risk factor related with increased carotid intima-media thickness and cardiovascular mortality [88]. In addition, whereas HDL is a powerful antioxidant, anti-inflammatory, and anti-atherogenic molecule, oxHDL is dysfunctional, loses its antiatherogenic benefits, and may potentially become a proinflammatory compound that may even contribute to the advancement of atherosclerosis [88,89,90].

Overall, there is compelling evidence that MA misuse and dependence produce damage due to oxidative stress and poorer antioxidant defenses, and that these processes play a crucial role in the enhanced neurotoxicity and atherosclerotic effects of MA.

### 4.3. MA Dependence, NOS Biomarkers and MIP

The third main finding of this study is that changes in OSTOX and ANTIOX biomarkers predict MA-induced PHEM symptoms and FTD, and that lowered HDL, zinc, and TAC and increased oxHDL and oxLDL are the most important predictors of psychotic symptoms. In addition, we found that MA dependence/dosing is substantially linked with psychotic symptoms and that the effects of MA usage on MIP are partly mediated by increased OSTOX and decreased ANTIOX biomarkers. About 30% of the variance in PHEM symptoms and FTD may be explained by the cumulative effects of oxLDL, decreased HDL, TAC, and zinc. Consequently, the substantial effects of MA dependency and MA dosage on MIP may be partially explained by enhanced neurotoxicity due to impacts on several pathways, such as increased oxidative toxicity, as described in the preceding section. Importantly, neurotoxicity with increased lipid and protein oxidation and decreased antioxidant defenses and associated immune pathways plays a significant role in schizophrenia, particularly in the more severe phenotypes [46,53,54]. In both MIP and severe schizophrenia, the OSTOX/ANTIOX ratio is raised, and increased OSTOX and decreased ANTIOX partially predict PHEM symptoms and FTD. Intriguingly, both MIP and paranoid schizophrenia are characterized by abnormalities in partial LINE 1 methylation, with the latter being more prominent in paranoid schizophrenia [25]. In addition, intertwined abnormalities in neuro-oxidative and neuro-immune pathways, and partial LINE 1 methylation may partially explain the pathogenesis of MIP and paranoid schizophrenia [25].

It is interesting to note that NO production was significantly decreased in the MA+PSO group and did not appear as a significant predictor of the symptoms in any of the multiple regression analyses. While increased NO production may have neurotoxic effects especially when oxygen radicals are increased, lowered levels of NO often indicate increased usage for formation of peroxynitrite, nitrosation and nitrosylation, which all have neurotoxic effects [91,92]. Future research should examine these NO-associated pathways in MIP.

Overall, the findings suggest that elevated OSTOX and decreased ANTIOX during acute MA intoxication are associated with MIP and that increased oxidative toxicity and lowered antioxidant defenses are shared pathways between MIP and schizophrenia. Nevertheless, our findings that “only” 29.9% of the variance in MA psychotic symptoms was explained by oxidative and antioxidant biomarkers indicates that a larger part of the variance is determined by other pathways.

## 5. Limitations

The present study would have been more interesting if we had examined the cytokine network, other oxidative stress biomarkers such as chlorinative stress, autoimmune responses to oxidative specific epitopes, and the tryptophan catabolite pathway, all of which play a role in schizophrenia [46,93]. It would have been interesting to assay glutathione in MIP patients because MA reduces glutathione levels including in the caudate of deceased patients [94]. It could be argued that the sample size is rather small. Nevertheless, a priori power analysis showed that the minimum number of participants should be 82 to obtain a power of 0.8, whilst post hoc analysis shows that the obtained power was 0.98. Due to cultural and religious restrictions, we were unable to include female Iraq patients with MA abuse. Therefore, our findings deserve replication in male and female patients in other cultures and countries. It could be argued that it is difficult to distinguish MIP from episodes of schizophrenia and new-onset schizophrenia. However, patients with a lifetime history of other axis-1 disorders, including schizophrenia and schizo-affective disorder, were excluded. In addition, all MIP patients showed complete remission of psychotic symptoms a few days (0.5–4) after acute intoxication, whereas schizophrenia is a chronic condition.

## 6. Conclusions

PLS analysis revealed that HDL, TAC, and zinc (all inversely) and oxLDL (positively) explained 29.9% of the variance in MIP severity (a first factor extracted from psychosis, excitation, mannerism, and formal thought disorder scores). The severity of MA dependence and MA dosing and route of administration predict 10.3% of the variance in HDL, 13.2% in oxLDL, 20.2% in TAC, and 15.2% in zinc. MA dependency and intoxication are associated with elevated oxidative stress and diminished antioxidant defenses, both of which enhance the risk of MIP during acute intoxication. On the basis of the data, it might be argued that antioxidant therapy may be effective for treating MIP. Nevertheless, given our new (to-be-submitted) results indicating that parts of the cytokine network are simply destroyed in young Thai MIP patients, we believe that the best approach to treat MIP is to quit taking MA and use new drugs that target aldehyde formation (now in phase 3) and to develop new treatments which try to improve the aberrations in immune functions.

It should be underscored that we included young patients with MA abuse and MIP and, therefore, future research should examine NOS and neuro-immune functions in older adults with MA use and MIP. Having observed these detrimental MA effects in younger patients, we anticipate observing even more disastrous neuro-immune and neuro-oxidative effects is older adults.

## Figures and Tables

**Figure 1 cells-11-03694-f001:**
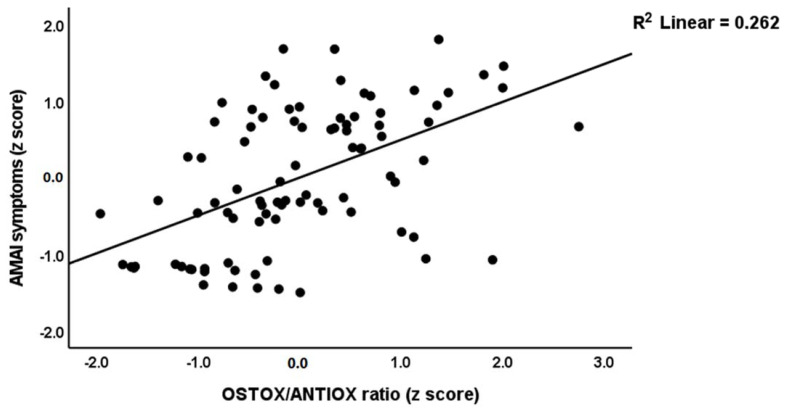
Partial regression plot of the acute methamphetamine intoxication (AMAI) symptom score on the oxidative stress toxicity/antioxidant (OSTOX/ANTIOX) ratio.

**Figure 2 cells-11-03694-f002:**
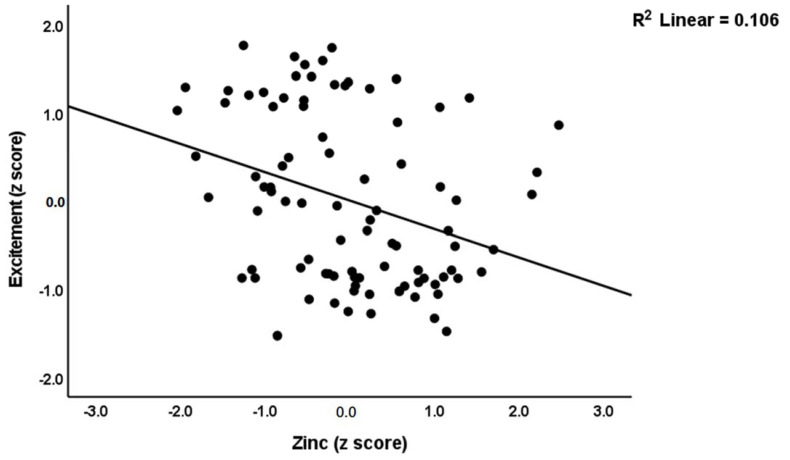
Partial regression plot of excitement on serum zinc levels.

**Figure 3 cells-11-03694-f003:**
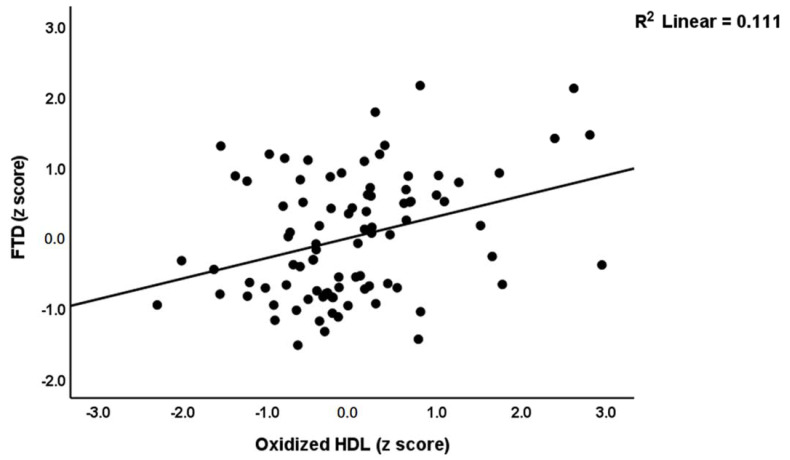
Partial regression plot of formal thought disorders (FTD) on serum oxidized high density lipoprotein (HDL) levels.

**Figure 4 cells-11-03694-f004:**
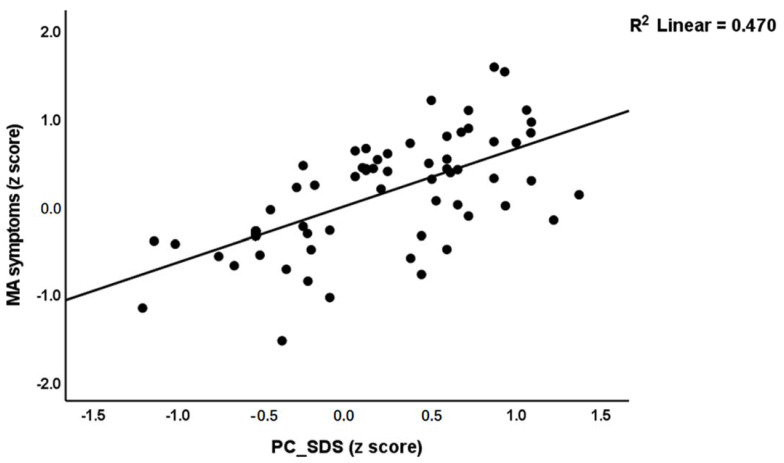
Partial regression plot of the overall severity of methamphetamine (MA)-induced psychotic symptoms (MA symptoms) on MA dependence (PC-SDS).

**Figure 5 cells-11-03694-f005:**
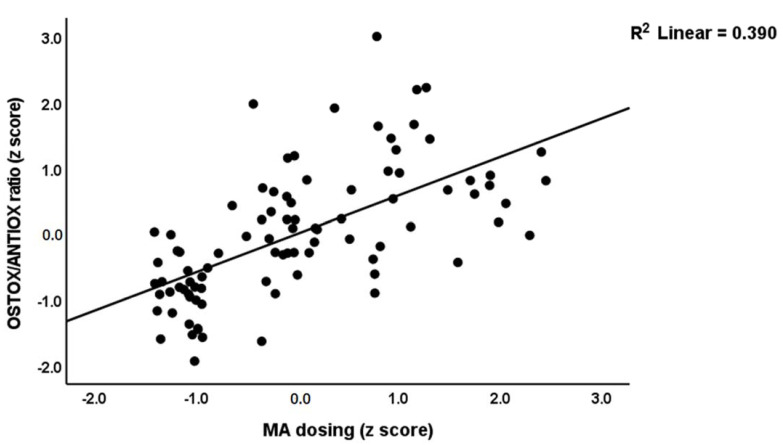
Partial regression plot of the oxidative toxicity/antioxidant defenses (OSTOX/ANTIOX) ratio on methamphetamine (MA) dosing.

**Figure 6 cells-11-03694-f006:**
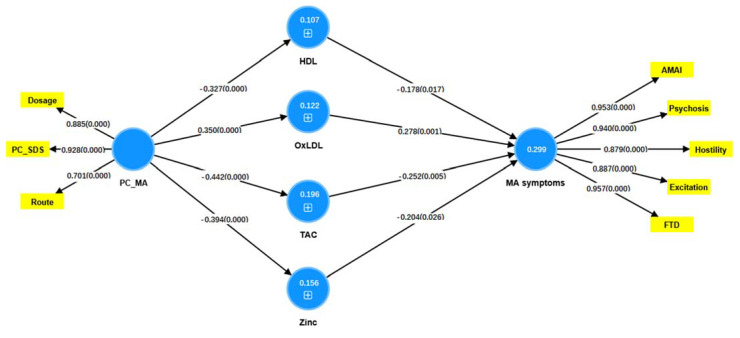
The results of the Partial Least Squares (PLS) analysis which show the impact of methamphetamine (MA) intake and dependence (PC_MA) on MA-induced symptoms which are partially mediated by increased levels of oxidized low-density lipoprotein (OxLDL), high-density lipoprotein (OxHDL) and lowered total antioxidant capacity (TAC) and zinc (Zn). MA symptoms are entered as a latent vector extracted from psychosis, hostility, excitation, mannerism, and MAI (acute MA-induced intoxication) symptoms. PC_MA is entered as a latent vector extracted from MA dosing, MA route of administration, and MA dependence (first principal component extracted from 4 SDS items). Path coefficients (with exact *p* values between brackets), loadings (with *p*-values) of the latent vectors and the explained variances (white figures in blue circles) are shown.

**Table 1 cells-11-03694-t001:** Socio-demographic and methamphetamine abuse (MA) data in healthy control participants (HCP) and MA patients classified as those with (MA+PSO) and without (MA-PSO) increased psychotic symptoms and oxidative stress biomarkers.

Variables	HCP (n = 30) ^A^	MA-PSO (n = 30) ^B^	MA+PSO (n = 30) ^C^	F/X^2^	df	*p*
Age (years)	27.3 (5.4)	24.4 (6.6) ^C^	28.6 (5.4) ^B^	3.99	2/87	0.022
BMI (Kg/m^2^)	25.33 (2.80)	25.24 (4.05)	24.94 (3.12)	0.11	2/87	0.894
Education (years)	10.9 (3.7)	7.8 (7.0)	10.2 (6.8)	2.16	2/87	0.121
Marital state (Single/Married)	8/22	13/17	8/22	2.54	2	0.280
Employment (No/Yes)	8/22 C	10/20 C	25/5 ^A,B^	23.07	2	<0.0001
Current MA use (No/Yes)	0/30	30/0	30/0	90.0		
PC_SDS (z score)	−1.370 (0.0) ^B,C^	0.611 (0.297) ^A,C^	0.758 (0.237) ^A,B^	KWT		<0.001
PC_MA dependence severity (z score)	−1.382 (0.0) ^B,C^	0.570 (0.166) ^A,C^	0.812 (0.219) ^A,B^	KWT		<0.0001
MA Administration route (O/S/I)	-	4/16/7	13/5/12	FFHET		0.001
Age at onset (years)	-	23.4 (5.9) ^C^	25.8 (4.90) ^B^	3.02	1/58	0.088
Duration of MA dependence (months)	-	14.8 (14.1) ^C^	34.9 (17.2) ^B^	24.54	1/57	<0.001
MA dosing (gm)	-	1.13 (0.47) ^C^	2.30 (0.79) ^B^	47.81	1/58	<0.001
Number of prior MIP episodes	-	2.0 (2.4)	2.2 (1.4)	0.07	1/58	0.798
Duration of the index MIP (days)	-	2.4 (2.3) ^A^	2.8 (2.0) ^A^	0.43	1/58	0.513
Days hospitalized due to MA intoxication	-	1.3 (1.7) ^A^	1.8 (1.8) ^A^	1.19	1/58	0.280
TUD (No/Yes)	15/15	7/23	1/29	17.29	2	<0.001
Alcohol dependence (No/Yes)	30/0	29/1	28/2	FFHET		0.770
Current drinker/past month (No/Yes)	30/0	29/1	28/2	FFHET		0.770
Lifetime Cannabis use (No/Yes)	30/0	29/1	28/2	FFHET		0.770
Any other substance, dependence	0	0	0	-	-	-

The results are shown as mean (SD) or as ratios: F: results of analysis of variance; X^2^: analysis of contingency tables. BMI: Body Mass Index, O/S/I: orally, smoking, injection, Kg: Kilogram, gm: Gram, TUD: Tobacco use disorder. ^A,B,C^: The results of pairwise comparisons among group means.

**Table 2 cells-11-03694-t002:** The clinical rating scales scores in healthy control participants (HCP) and patients with methamphetamine abuse (MA) classified into those with (MA+PSO) and without (MA-PSO) increased psychotic symptoms and oxidative stress (PSO).

Variables	HCP (n = 30) ^A^	MA-PSO (n = 30) ^B^	MA+PSO (n = 30) ^C^	F	df	*p*
MAI symptoms (z score)	−1.052 (0.077) ^B,C^	−0.114 (0.080) ^A,C^	1.165 (0.078) ^A,B^	208.77	2/85	<0.0001
Psychosis (z score)	−1.190 (0.090) ^B,C^	0.296 (0.094) ^A,C^	0.894 (0.091) ^A,B^	143.17	2/85	<0.0001
Hostility (z score)	−1.085 (0.112) ^B,C^	0.317 (0.117) ^A,C^	0.769 (0.114) ^A,B^	74.58	2/85	<0.0001
Excitement (z score)	−1.038 (0.115) ^B,C^	0.212 (0.120) ^A,C^	0.827 (0.116) ^A,B^	69.03	2/85	<0.0001
Mannerism (z score)	−0.562 (0.162) ^B,C^	0.122 (0.169) ^A^	0.440 (0.164) ^A^	10.06	2/85	<0.001
Formal though disorders (z score)	−1.143 (0.072) ^B,C^	0.036 (0.074) ^A,C^	1.107 (0.072) ^A,B^	250.22	2/85	<0.0001

All results of univariate GLM analysis; data are expressed as mean (SE), i.e., estimated marginal means obtained by GLM analysis after covarying for age and education. ^A,B,C^: The results of pairwise comparisons among group means. MAI: most prominent MA intoxication symptom score, based on delusions, conceptual disorganization, suspiciousness, and difficulties in abstract thinking.

**Table 3 cells-11-03694-t003:** Oxidative stress biomarkers in heathy control participants (HCP) and patients with methamphetamine abuse (MA) classified into those with (MA+PSO) and without (MA-PSO) increased psychotic symptoms and oxidative stress (PSO).

Variables	HCP (n = 30) ^A^	MA-PSO (n = 30) ^B^	MA+PSO (n = 30) ^C^	F	df	*p*	Partial Eta Squared
Catalase (ng/mL)	4.76 (0.30) ^C^	4.26 5(0.31)	3.73 (0.31) ^A^	2.91	2/85	0.060	0.064
GPx (U/mL)	19.58 (1.39) ^C^	22.14 (1.43) ^C^	15.28 (1.42) ^A,B^	5.71	2/85	0.005	0.118
Myeloperoxidase ^#^ (U/L)	110.16 (5.74)	115.70 (5.90)	125.30 (5.84)	2.64	2/85	0.077	0.058
Malondialdehyde (nM)	1213.2 (68.8) ^B,C^	1590.4 (70.6) ^A^	1494.9 (70.0) ^A^	8.03	2/85	<0.001	0.159
OxHDL (U/mL)	136.1 (8.3) ^C^	152.4 (8.5)	174.7 (8.5) ^A^	5.35	2/85	0.006	0.112
OxLDL (ng/mL)	50.1 (3.7) ^B,C^	67.3 (3.8) ^A^	65.8 (3.7) ^A^	6.77	2/85	0.002	0.137
TAC (U/mL)	4.36 (0.25) ^B,C^	3.42 (0.26) ^A,C^	2.53 (0.26) ^A,B^	13.36	2/85	<0.001	0.239
NO (uM)	29.44 (1.44) ^C^	32.67 (1.48) ^C^	24.92 (1.47) ^A,B^	6.68	2/85	0.002	0.136
Zinc (mg/l)	0.761(0.025) ^C^	0.704 (0.026) ^C^	0.596 (0.026) ^A,B^	10.94	2/85	<0.001	0.205
HDL (mM)	1.203 (0.023) ^B,C^	1.130 (0.024) ^A^	1.111 (0.024) ^A^	4.39	2/85	0.015	0.094
OSTOX (z score)	−0.710(0.159) ^B,C^	0.225(0.163) ^A^	0.485(0.162) ^A^	15.64	2/85	<0.001	0.269
ANTIOX (z score)	0.646(0.151) ^B,C^	0.120(0.155) ^A,C^	−0.766(0.154) ^A,B^	21.93	2/85	<0.001	0.340
OSTOX/ANTIOX (z score)	−0.886(0.134) ^B,C^	0.069(0.138) ^A,C^	0.817(0.136) ^A,B^	40.39	2/85	<0.001	0.487

All results of univariate GLM analysis; data are expressed as mean (SE), i.e., estimated marginal means obtained by GLM analysis after covarying for age and BMI. ^#^: Processed in Logarithm transformation. MPO: Myeloperoxidase, TAC: Total antioxidant capacity, CAT: Catalase, GPx: Glutathione peroxidase, MDA: Malondialdehyde, OxHDL: Oxidized High-density lipoprotein, OxLDL: Oxidized Low-density lipoprotein, NO: Nitric oxide, Zn: Zinc, OSTOX: Index of oxidative stress, ANTIOX: Index of antioxidant defenses. ^A,B,C^: The results of pairwise comparisons among group means.

**Table 4 cells-11-03694-t004:** Intercorrelations between severity of methamphetamine (MA) dependence (PC_SDS) and severity of MA dependence (PC_MA), methamphetamine abuse (MA), oxidative stress parameters, and symptoms domains.

Variables	All Subjects Combined (n = 90)					MA Dependence (n = 60)	
	PC_SDS	PC_MA	OSTOX	ANTIOX	OSTOX/ANTIOX	PC_SDS	PC_MA
MAI symptoms	0.783 **	0.799 **	0.434 **	−0.415 **	0.555 **	0.330 *	0.483 **
Psychosis	0.873 **	0.871 **	0.491 **	−0.402 **	0.583 **	0.498 **	0.524 **
Hostility	0.790 **	0.793 **	0.360 **	−0.285 *	0.422 **	0.189	0.201
Excitement	0.782 **	0.779 **	0.336 **	−0.269 *	0.396 **	0.409 **	0.428 **
Mannerism	0.453 **	0.477 **	0.220 *	−0.109	0.215 *	0.110	0.075
Formal Thought Disorders	0.839 **	0.850 **	0.451 **	−0.426 **	0.573 **	0.382 **	0.500 **
OSTOX	0.520 **	0.520 **	-			0.164	0.168
ANTIOX	−0.443 **	−0.482 **				0.049	−0.169
OSTOX/ANTIOX ratio	0.629 **	0.654 **				0.080	0.243

* *p* < 0.05; ** *p* < 0.001. MAI: most prominent MA-intoxication symptoms, based on delusions, conceptual disorganization, suspiciousness, and difficulties in abstract thinking. OSTOX: index of oxidative toxicity; ANTIOX: index of antioxidant defenses.

**Table 5 cells-11-03694-t005:** The results of multiple regression analyses with the methamphetamine (MA) intoxication symptoms as dependent variables and oxidative stress biomarkers as explanatory variables.

Dependent Variables	Explanatory Variables	Parameter Estimates + Statistics			Model Statistics and Effect Size			
		β	t	*p*	R^2^	F	df	*p*
#1. MAI symptoms	**Model **				0.352	9.13	5/84	<0.001
	TAC	−0.220	−2.31	0.023				
	OxHDL	0.276	3.04	0.003				
	HDL	−0.187	−2.05	0.043				
	Zinc	−0.229	−2.36	0.020				
	OxLDL	0.191	2.09	0.040				
#2. MAI symptoms	**Model **				0.308	39.11	1/88	<0.001
	OSTOX/ANTIOX	0.555	6.25	<0.001				
#3 Psychosis	**Model **				0.316	13.23	3/86	<0.001
	TAC	−0.241	−2.51	0.014				
	OxLDL	0.345	3.82	<0.001				
	Zinc	−0.273	−2.88	0.005				
#4. Hostility	**Model **				0.147	7.49	2/87	<0.001
	TAC	−0.264	−2.63	0.010				
	OxLDL	0.243	2.43	0.017				
#5. Excitement	**Model **				0.175	6.07	3/86	<0.001
	Zinc	−0.322	−3.19	0.002				
	OxLDL	0.268	2.71	0.008				
	Age	0.221	2.17	0.032				
#6. Mannerism	**Model **				0.141	7.12	2/87	0.001
	OxLDL	0.309	3.11	0.003				
	Zinc	−0.219	−2.21	0.030				
#7. Formal thought disorders	**Model **				0.355	9.25	5/84	<0.001
	TAC	−0.265	−2.78	0.007				
	OxHDL	0.261	2.88	0.005				
	MPO	0.190	2.12	0.037				
	Zinc	−0.242	−2.56	0.012				
	OxLDL	0.184	1.99	0.049				

MAI: MA-induced intoxication symptoms. TAC: Total antioxidant capacity, OxHDL: Oxidized High-density lipoprotein, OxLDL: Oxidized Low-density lipoprotein, OSTOX: Index of oxidative stress, ANTIOX: Index of antioxidant defenses, OSTOX/ANTIOX ratio: zzOSTOX-zANTIOX composite score, OxLDL: Oxidized Low-density lipoprotein, TAC: Total antioxidant capacity, OxHDL: Oxidized High-density lipoprotein, MPO: Myeloperoxidase.

**Table 6 cells-11-03694-t006:** The results of multiple regression analyses with methamphetamine (MA)-induced symptoms and the oxidative stress toxicity/antioxidant defenses (OSTOX/ANTIOX) ratio as dependent variables and MA dependence and MA intake features as explanatory variables.

Dependent Variables	Explanatory Variables	Parameter Estimates + Statistics			Model Statistics and Effect Size			
		β	t	*p*	R^2^	F	df	*p*
#1 MA symptoms	**Model **				0.818	194.96	2/87	<0.001
	PC_SDS	0.630	8.79	<0.001				
	MA dosing	0.324	4.53	<0.001				
#2. OSTOX/ANTIOX ratio	**Model **				0.447	35.18	2/87	<0.001
	PC_SDS	0.449	3.60	<0.001				
	MA dosing	0.258	2.07	0.041				

PC_SDS: a factor reflecting severity of MA dependence. MA dosing: the dosing of MA prior to hospital admission.

## Data Availability

The dataset generated during and/or analyzed during the current study will be available from the corresponding author (M.M.) upon reasonable request and once the dataset has been fully exploited by the authors.

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
