# Peer review of "Increased Lipid Peroxidation and Lowered Antioxidant Defenses Predict Methamphetamine Induced Psychosis"

_cells, 2022, doi:10.3390/cells11223694_

Round 1

Reviewer 1 Report

Dear Authors:

The manuscript "Increased lipid peroxidation and lowered antioxidant defenses predict methamphetamine induced psychosis" by Kadhem Al-Hakeim et al has demonstrated MA dependence and intoxication are associated with increased oxidative stress and lowered antioxidant defenses, which both increase risk of MIP during acute intoxication. MA dependence is accompanied by increased atherogenicity due to lowered HDL and increased oxLDL and oxHDL. I have just a few suggestions.

1. Some background information or references are missing. In introduction, please add more background information about the role of ROS in disease development, which can emphasize the importance of your article. Some reviews have summarized it. (please cite: 1. Advances in the Prevention and Treatment of Obesity-Driven Effects in Breast Cancers. Front Oncol. 2022 Jun 22;12:820968. doi: 10.3389/fonc.2022.820968. PMID: 35814391; PMCID: PMC9258420.

2. An Epigenetic Role of Mitochondria in Cancer. Cells. 2022 Aug 13;11(16):2518. doi: 10.3390/cells11162518. PMID: 36010594; PMCID: PMC9406960.

3. Mitochondrial mutations and mitoepigenetics: Focus on regulation of oxidative stress-induced responses in breast cancers. Semin Cancer Biol. 2022 Aug;83:556-569. doi: 10.1016/j.semcancer.2020.09.012. Epub 2020 Oct 6. Erratum in: Semin Cancer Biol. 2022 Jul 16;: PMID: 33035656.)

2. The manuscript needs linguistic improvement.

Best, 

Author Response

The manuscript "Increased lipid peroxidation and lowered antioxidant defenses predict methamphetamine induced psychosis" by Kadhem Al-Hakeim et al has demonstrated MA dependence and intoxication are associated with increased oxidative stress and lowered antioxidant defenses, which both increase risk of MIP during acute intoxication. MA dependence is accompanied by increased atherogenicity due to lowered HDL and increased oxLDL and oxHDL. I have just a few suggestions.

  1. Some background information or references are missing. In introduction, please add more background information about the role of ROS in disease development, which can emphasize the importance of your article. Some reviews have summarized it. (please cite: 1. Advances in the Prevention and Treatment of Obesity-Driven Effects in Breast Cancers. Front Oncol. 2022 Jun 22;12:820968. doi: 10.3389/fonc.2022.820968. PMID: 35814391; PMCID: PMC9258420.
  2. An Epigenetic Role of Mitochondria in Cancer. Cells. 2022 Aug 13;11(16):2518. doi: 10.3390/cells11162518. PMID: 36010594; PMCID: PMC9406960.
  3. Mitochondrial mutations and mitoepigenetics: Focus on regulation of oxidative stress-induced responses in breast cancers. Semin Cancer Biol. 2022 Aug;83:556-569. doi: 10.1016/j.semcancer.2020.09.012. Epub 2020 Oct 6. Erratum in: Semin Cancer Biol. 2022 Jul 16;: PMID: 33035656.)
  4. The manuscript needs linguistic improvement.

@@RESPONSE: Our paper is about MA-induced psychosis and not breast cancer. If there had been any connection between MA use and breast cancer, I could have discussed this connection, and the underlying NOS mechanisms. However, there are no reports on any association between MA use and breast cancer. Thus, I cannot cite the papers proposed here.

Reviewer 2 Report

In this manuscript Al-Hakeim et al. used machine learning methods to study the role of oxidative/nitrosative stress on the severity of the symptoms in people with methamphetamine (METH) induced psychosis. Through psychiatric tests and serum detection of antioxidant and oxidant molecules, the authors found that METH-intoxicated patients have increased oxidative/nitrosative levels in comparison with a control group, and, oxidative stress levels are directly proportional in relationship with the psychiatric symptoms’ severity. However, all the oxidative/nitrosative stress-related biomarkers only explained around 30% of the variance of the METH-induced psychotic symptoms.

This is an interesting study that uses precision nomothetic medicine approaches for a better understanding of the oxidative and nitrosative stress phenomena in METH-induced psychosis, which can be useful for other substance use disorders.

I have some concerns, comments, and suggestions which are listed below.

Major

·       The authors state that only males were included in the study and provide reasonable statements for this. However, there is no discussion about how the results may be biased because of this. Also, only male inclusion is not considered a limitation of the study, please explain this.

·       The first episode of psychosis (FEP) commonly appears in early adulthood. Also, FEP patients have increased oxidative/nitrosative stress levels in peripheral blood, and it is known that all the quantified oxidative/nitrosative stress-related molecules in this study (patients with METH-induced psychosis) are altered similarly in FEP. Is there a possibility that the group with severe symptoms and high oxidative/nitrosative stress levels was people with FEP? How can it be differentiated between the patients with METH-induced psychosis and the FEP patients? If these questions are unable to answer it should be discussed this issues and added to the limitations of the study.

·       It is stated that people with affective disorders were discarded from the study. Does anxiety was included in this criterion? Anxiety can boost oxidative molecules and may be contributing to bias. Does an anxiety test was performed on the included patients? If it was done it can be included the anxiety effect on the evaluated parameters.

Minor

·       I consider it is important to highlight the age of the studied population, since METH consumption increases in the adolescent population. Thus, I suggest stating that this study was carried out in the early-adulthood population.

·       Please provide the following abbreviation meanings throughout the manuscript:

o   LINE 1

o   LSD test

o   MAI symptoms

o   PHEM symptoms

Author Response

Major

  • The authors state that only males were included in the study and provide reasonable statements for this. However, there is no discussion about how the results may be biased because of this. Also, only male inclusion is not considered a limitation of the study, please explain this.

@@RESPONSE: This is now discussed in the limitations section. It reads:

Due to cultural and religious restrictions, we were unable to include female patients with MA abuse. Therefore, our findings deserve replication in male and female patients in other cultures and countries.

  • The first episode of psychosis (FEP) commonly appears in early adulthood. Also, FEP patients have increased oxidative/nitrosative stress levels in peripheral blood, and it is known that all the quantified oxidative/nitrosative stress-related molecules in this study (patients with METH-induced psychosis) are altered similarly in FEP. Is there a possibility that the group with severe symptoms and high oxidative/nitrosative stress levels was people with FEP? How can it be differentiated between the patients with METH-induced psychosis and the FEP patients? If these questions are unable to answer it should be discussed this issues and added to the limitations of the study.

RESPONSE: addressed in the text as:

It could be argued that it is difficult to distinguish MIP from episodes of schizophrenia and new-onset schizophrenia. However, patients with a lifetime history of other axis-1 disorders, including schizophrenia and schizo-affective disorder, were excluded. In addition, all MIP patients showed complete remission of psychotic symptoms a few days after the acute intoxication, whereas schizophrenia is a chronic condition.

  • It is stated that people with affective disorders were discarded from the study. Does anxiety was included in this criterion? Anxiety can boost oxidative molecules and may be contributing to bias. Does an anxiety test was performed on the included patients? If it was done it can be included the anxiety effect on the evaluated parameters.

@@RESPONSE: people with the major anxiety disorders were excluded. This is now specified in the M&M section:

Patients were excluded if they showed a lifetime or current diagnosis of other axis-1 diagnoses including mood disorders, schizophrenia, schizo-affective psychoses, obsessive compulsive disorder, post-traumatic stress disorder, generalized anxiety disorder, panic disorder, autism spectrum disorders.

Furthermore, the anxiety that may accompany acute MIP is not likely to cause increased aldehyde formation, as this is a more chronic process.

Minor

  • I consider it is important to highlight the age of the studied population, since METH consumption increases in the adolescent population. Thus, I suggest stating that this study was carried out in the early-adulthood population.

@@RESPONSE: In the Abstract, we now underscore that we recruited younger males (see Intro please). P.S. I use “younger”, as not all were in their early adulthood (which is between 18-25)

  • Please provide the following abbreviation meanings throughout the manuscript:

@@RESPONSE: These are now spelled out in the text with abbreviation and we have added a list with abbreviations.

o   LINE 1

o   LSD test

o   MAI symptoms

  • PHEM symptoms

Reviewer 3 Report

The authors present an interesting study that seeks to determine which factors contribute to predicting the onset of psychosis in people dependent on methamphetamines. They evaluate a series of oxidative stress parameters and serum antioxidants to do this. Through various statistical analyses, they characterize two population clusters and propose a prediction model. However, several issues must be taken into account before publication.

1. There are many abbreviations in the manuscript; having a list of them would be appreciated.

2. In the M&M biomarkers assays section, the authors indicate the composition of three scores; the "c" is missing.

3. In the statistical analysis, the paragraph about partial least squares "alterations in ONS biomarkers" is mentioned, although they are referred to as NOS in the rest of the manuscript. Another abbreviation whose meaning should be explained at its first mention is PSO and PHEM symptoms.

4. In the demographic table of the participating patients, it is observed that they have had at least two previous episodes of MA-induced psychosis, so the authors should indicate if they are currently consuming any antipsychotic (AP) or are free of medication. If there is a consumption of AP, it should be indicated and discussed since some AP have anti-inflammatory and antioxidant effects and could affect the results.

5. The study population is young; are these findings reproducible in the adult population? What happens with the organism's plasticity to respond to this oxidative imbalance with age?

6. Among the antioxidant factors, the most important and robust in several psychiatric pathologies is glutathione; why has it not been considered for these analyses? Are there previous related studies?

7. How is the response to treatment in these two clusters of patients? are there differences?

8. What future therapeutic prospects could be considered for these patients? The use of anti-inflammatory/antioxidant adjuvants has been proposed in several psychiatric diseases with controversial results.

9. Alterations in the inflammatory/oxidant balance have already been proposed in several psychiatric pathologies; this prediction model could be extrapolated to these diseases? Or only predicts this specific setting in the youth population.

10. Finally, emphasize the critical limitation of this study by not having women; the authors explain this fact but do not include it in the limitations section.

Author Response

The authors present an interesting study that seeks to determine which factors contribute to predicting the onset of psychosis in people dependent on methamphetamines. They evaluate a series of oxidative stress parameters and serum antioxidants to do this. Through various statistical analyses, they characterize two population clusters and propose a prediction model. However, several issues must be taken into account before publication.

  1. There are many abbreviations in the manuscript; having a list of them would be appreciated.

@RESPONSE: we have added a list with abbreviations.

  1. In the M&M biomarkers assays section, the authors indicate the composition of three scores; the "c" is missing.

@@RESPONSE: is added

  1. In the statistical analysis, the paragraph about partial least squares "alterations in ONS biomarkers" is mentioned, although they are referred to as NOS in the rest of the manuscript. Another abbreviation whose meaning should be explained at its first mention is PSO and PHEM symptoms.

@@RESPONSE: NOS all over the text.

  1. In the demographic table of the participating patients, it is observed that they have had at least two previous episodes of MA-induced psychosis, so the authors should indicate if they are currently consuming any antipsychotic (AP) or are free of medication. If there is a consumption of AP, it should be indicated and discussed since some AP have anti-inflammatory and antioxidant effects and could affect the results.

@@RESPONSE: Not one of the patients was treated with antipsychotic agents prior to hospitalization. Now specified in the text as:

Prior to hospitalization and blood sampling, no antipsychotic medications were administered to any of the patients. All patients were admitted to the hospital for the first time.

  1. The study population is young; are these findings reproducible in the adult population? What happens with the organism's plasticity to respond to this oxidative imbalance with age?

@@RESPONSE: this is addressed in the limitation section as:

It should be underscored that we included young patients with MA abuse and MIP and, therefore, future research should examine NOS and neuro-immune functions in older adults with MA use and MIP. Having observed these detrimental effects in younger patients, we anticipate observing even more disastrous neuro-immune and neuro-oxidative effects in older adults.

  1. Among the antioxidant factors, the most important and robust in several psychiatric pathologies is glutathione; why has it not been considered for these analyses? Are there previous related studies?

@@RESPONSE: Addressed in the limitations section as:

It would have been interesting to assay glutathione in our MIP patients because MA reduces glutathione levels including in the caudate of deceased patients (Uys et al., 2014)

  1. How is the response to treatment in these two clusters of patients? are there differences?

@@RESPONSE. Very difficult to examine with statistical tests because the MIPs we observed were always short-lasting psychoses and all MIP symptoms disappeared within a few days (i.e. 0.5-4 days).

  1. What future therapeutic prospects could be considered for these patients? The use of anti-inflammatory/antioxidant adjuvants has been proposed in several psychiatric diseases with controversial results.

@@RESPONSE: This is addressed at the end of the paper as:

On the basis of the data, it might be argued that antioxidant therapy may be effective for treating MIP. Nevertheless, given our new (to-be-submitted) results indicating that parts of the cytokine network are simply destroyed in young Thai MIP patients, we believe that the best approach to treat MIP is to quit taking MA and use new drugs that target aldehyde formation and to develop new treatments which try to improve the aberrations in immune functions.

Thus, MIP is not only associated with severe lipid peroxidation and in addition with major (in fact it is terrible) aberrations in the cytokine network. I never saw such aberations in the cytokine network in any disease.

  1. Alterations in the inflammatory/oxidant balance have already been proposed in several psychiatric pathologies; this prediction model could be extrapolated to these diseases? Or only predicts this specific setting in the youth population.

@@RESPONSE: MIP is often used as a model of schizophrenia but this is a bridge too far. Let me discuss this in my forthcoming paper on the neuro-immune effects of MA in MIP (a complete disaster for the patients !!!). But: the immune profile is completely different between MA-use/MIP and schizophrenia. P.S. We now write up these results. It is too early to disclose now.

  1. Finally, emphasize the critical limitation of this study by not having women; the authors explain this fact but do not include it in the limitations section.

@@RESPONSE: this is addressed as:

Due to cultural and religious restrictions, we were unable to include female patients with MA abuse. Therefore, our findings deserve replication in male and female patients in other cultures and countries.